# A Path-Based Selection Solution Approach for the Low Carbon Vehicle Routing Problem with a Time-Window Constraint

**Xianlong Ge** [1] , **Xiaobo Ge** [1] **and Weixin Wang** [2,*]

1   Economics and Management School, Chongqing Jiaotong University, Chongqing 400074, China;
    gexianlong@cqjtu.edu.cn (X.G.); gexb@lzlj.com (X.G.)
2   Research Centre for International Business and Economics, International Business School, Sichuan
    International Studies University, Chongqing 400031, China
*   Correspondence: wangweixin@sisu.edu.cn

**Abstract:** Due to the gradual improvement of urban traffic network construction and the increasing number of optional paths between any two points, how to optimize a vehicle travel path in a multi-path road network and then improve the efficiency of urban distribution has become a difficult problem for logistics companies. For this purpose, a mixed-integer mathematical programming model with a time window based on multiple paths for urban distribution in a multi-path environment is established and its exact solution solved using software CPLEX. Additionally, in order to test the application and feasibility of the model, simulation experiments were performed on the four parameters of time, distance, cost, and fuel consumption. Furthermore, using Jingdong (JD), the main urban area in Chongqing, as an example, the experimental results reveal that an algorithm that considers the path selection can significantly improve the efficiency of urban distribution in metropolitan areas with complex road structures.

**Keywords:** urban transportation; multiple paths; vehicle routing problem

## 1. Introduction

In the past few decades, with the increasing development of China's urbanization process, traffic congestion, especially in large cities, has become increasingly serious, which has attracted scholars' extensive attention. In order to solve this serious problem, many cities have increased their investment in transportation infrastructure, which makes city roads more intricate and provides more alternative paths for urban distribution. In the traditional pollution emissions problem (PEP) solving process, there is just one route between two customers. The aim of optimization is to determine the customer access order of vehicles in an urban distribution with the goal of minimizing emissions. In these classic pollution routing problems (PRPs), suboptimal alternative paths are eliminated, leaving only one path between two client nodes. Indeed, the quality sequence of the roads will change as time goes by. It is important to consider alternative paths in the new condition. The multi-path vehicle routing problem studied in this paper adds alternative paths into the urban distribution, which makes the vehicle always choose other smoother paths when it encounters traffic jams during distribution, most of which may not be the shortest path. However, the distribution cost of the vehicles is even smaller because delivery vehicles avoid road congestion. Therefore, how to choose the appropriate travel route and then minimize possible travel congestion in the complex road network has become the main challenge logistics companies face.

With the continuous improvement of urban transportation network construction, the optionality of urban roads has been enhanced, which provides a new approach for solving urban congestion.

In this condition, an increasing number of scholars have begun to pay more attention to the optionality of paths and have studied road networks showing multi-graph and path flexibility. Garaix et al. [1] studied the multi-attribute vehicle routing problem and proposed a multi-graph representation of the road network. Setak et al. [2] solved the time-dependent vehicle path problem in a multi-graph with first-in and first-out (FIFO) properties through a heuristic tabu search (TS) algorithm. Demir et al. [3] first proposed the pollution emissions problem (PEP), which primarily used the integrated modal emissions model (IMEM) to calculate the vehicle's fuel consumption. Ehmke et al. [4] studied the issue of emissions-minimized vehicle routing with time-dependence. Grote et al. [5] extended PRP to dual-objective PRP, the objective function of which aims to reduce fuel consumption and travel time. In order to minimize travel and fixed costs, Koç et al. [6] considered the PRP of heterogeneous vehicles and determined the number of each vehicle type and the driving route of these vehicles. Konak and Xiao. [7] considered emission minimization in the problem of location routing of a heterogeneous fleet and divided the city into different regions, each with a constant (time-independent) speed. When the vehicle travels between different areas with different loads, it can choose different paths between areas to minimize emissions. Barth and Boriboonsomsin [8] established a time-dependent network model that relies on FIFO attributes. Ichoua et al. [9] proposed a model with a step function for driving speed and a piecewise linear function for vehicle travel time. This method has been widely used in other studies. Kim et al. [10] used a heuristic algorithm to solve the time-dependent vehicle routing problem (TDVRP) in dynamic vehicle routing problems and reported that using the time-dependent shortest path in TDVRP can significantly reduce vehicle travel time. Bektas and Laporte [11] analyzed the pollution routing problem based on emission and energy consumption models; moreover, the effects of time windows, speed, distance and other factors on vehicle emissions were considered. Repoussis et al. [12] studied the open vehicle routing problem with time windows. Wang et al. [13] considered the impact of ramp factors on emissions in the vehicle routing problem (VRP) and proposed a two-objective strategy for energy consumption minimizing low-carbon Vehicle routing problems (ECM-LCVRP) in different road gradient environments. Based on the classical TDVRP, Liu and Zhang. [14] constructed a model of the urban distribution problem that comprehensively considered energy conservation, low carbon and cost saving and further minimized economic cost, including the above three factors; the goal was to plan the vehicle routing problem. On the basis of studying the vehicle fuel consumption model, time-window penalty function and speed optimization strategy, Ge et al. [15] proposed a variable-speed vehicle routing optimization model with a time window so as to solve the problem of the difficulty of a vehicle traveling with constant speed to meet the time-to-service requirement of its customers. A low-carbon pickup and delivery vehicle routing problem was proposed by Qin et al. [16], the adaptive genetic hill-climbing algorithm was designed to solve the optimization model, which considers the carbon tax policy. Bravo et al. [17] analyzed the pickup and delivery vehicle pollution routing problem with multi-objectives, and the total traveling time, the emission of greenhouse gases and the number of customers were considered in the model. An evolutionary algorithm was designed to solve this problem. The multi-objective regional low-carbon location routing problem was proposed by Leng, L.L. [18]. The total cost, time and service duration were considered in the model, three multi-objective evolutionary algorithms were designed based on the complexity of the proposed problem. Shen, L., et al. [19] described an open vehicle routing problem with time windows, and the low-carbon open vehicle routing problem with time-windows model was established, and the goal was minimum total costs. A two-phase algorithm was designed to handle the model. Niu, Y.Y, et al. [20] analyzed the green open vehicle routing problem with time windows. The comprehensive modal emission model (CMEM) was established, and a hybrid tabu search algorithm with several neighborhood search strategies was designed to handle this problem.

　　Based on the analysis above, it can be concluded that the former research on vehicle routing problems with time windows (VRPTW) and PEP focused on the factors of traffic congestion, vehicle composition and vehicle load, etc., rather than multi-path vehicle routing problems [21–23]. Therefore, a multi-path mixed-integer mathematical programming model with a time window was established,

which aimed to optimize fuel cost, driver cost, vehicle depreciation and time-window penalty cost and determine the exact solution using CPLEX in Java. Finally, the applicability and feasibility of the model were verified with the example of Jingdong's (JD) logistics in the main urban area of Chongqing, and sensitivity analyses were performed on this model.

## 2. Problem Description

The problems studied in this paper can be described as follows: distribution centers offer services to customers scattered in urban areas through a group of homogeneous fleets. Delivery is limited by the number of vehicles, vehicle capacity, customer service time and routing selection. The purpose is to minimize the sum of fuel consumption cost, vehicle depreciation cost and driver's salary by flexibly selecting the driving route while meeting the requirements of customer needs and vehicle capacity. Fuel consumption, at this point, depends largely on the vehicle's speed, load and distance, while drivers are paid from the start of the vehicle until it returns to its starting point.

### 2.1. Multi-Path Path Selection

The traffic conditions of urban road networks have significant differences in time and space. Therefore, there are multiple paths between different nodes $(i, j)$ to choose within an actual urban distribution network. Depending on the road conditions, delivery vehicles, in this way, can flexibly select the travel route to avoid urban congestion. In this paper, an optional path (OP) is defined as $G(N, H)$, a multi-path network, where node N represents a collection of customer and depot locations, and H represents a collection of road sections in a route formulation. Figure 1 shows an n-nodes graph with multiple edges to represent the path conditions under multiple paths.

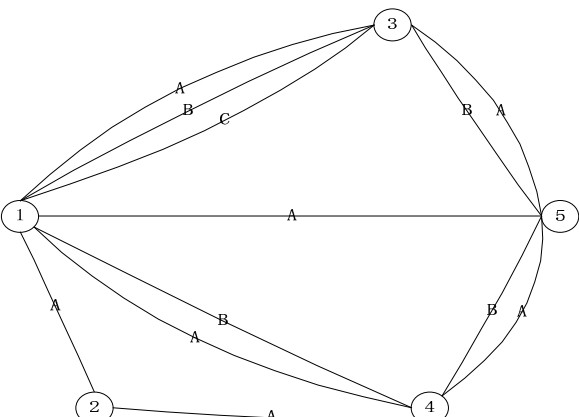

**Figure 1.** Schematic diagram of multi-path network.

Node 1 is the depot location, and the remaining are customer nodes; $w_{i,j}$ is the total number of edges between nodes $(i, j)$. For example, $w_{3,5} = 2$ indicates that there are two edges between nodes (3,5), that is, two existing road connections between them. As seen in Figure 1, multiple optional paths exist in any two nodes. According to the real-time data of the traffic information management system, logistics companies can obtain real-time traffic information and then use it to select the appropriate travel route. There may be multiple paths between any two points in Figure 1 because of many edges existing between them (each edge represents an optional travel path), one of which has time, distance, cost and fuel consumption properties. In this paper, our goal is to find the optimal travel route under different conditions based on the capacity and time constraints and then to meet the distribution needs of different logistics companies.

## 2.2. The Adjacency-Matrix Representation of a Multi-Path

Since the number of paths between any two nodes may provide many choices, this undoubtedly makes the relevant data too large to deal with [24,25]. Therefore, in this paper, methods were first performed to decrease the dimensions of the pre-collected traffic data, making the next steps more convenient. Firstly, all possible travel paths were represented by three $n*n$ matrices. Next, the time, speed, distance, fuel consumption and cost values of the different labeled paths A, B and C, respectively, were plugged into the above matrix, where the values of distance and time were derived from the traffic information management system, the fuel consumption was calculated by Equations (5) and (6) and the cost values were calculated by Equations (7)–(10). The driving speed on the arc $(i, j)$ was $s_{ij} = d_{ij}/t_{ij}$; the specific matrix data are shown in the Appendix A, Table A1. Finally, through the Formulas (1)–(4), operating the matrix of time, velocity, distance, fuel consumption and cost matrix, respectively, with the different path labels A, B and C, the minimum distance matrix, time matrix, fuel consumption matrix and cost matrix in all the graphs could be obtained separately, which could degrade the multi-path problem into a deterministic path problem, which was advantageous for the next calculation and solution.

$$C_{dis} = \min\left\{C_{dis}^A C_{dis}^B C_{dis}^C\right\} \tag{1}$$

$$C_{time} = \min\left\{C_{time}^A C_{time}^B C_{time}^C\right\} \tag{2}$$

$$C_{fuel} = \min\left\{C_{fuel}^A C_{fuel}^B C_{fuel}^C\right\} \tag{3}$$

$$C_{cost} = \min\left\{C_{cost}^A C_{cost}^B C_{cost}^C\right\} \tag{4}$$

There are three paths (A, B, C) between two points, and the shortest distance among the three paths is represented by $C_{dis}$, $C_{dis}^A$, $C_{dis}^B$, $C_{dis}^C$ represent the distances of paths A, B and C respectively. $C_{time}$ represent the shortest time, $C_{time}^A$, $C_{time}^B$, $C_{time}^C$ represent the time of paths A, B and C respectively. $C_{fuel}C_{time}$ represent the minimum fuel consumption, $C_{fuel}^A$, $C_{fuel}^B$, $C_{fuel}^C$ represent the fuel consumption of paths A, B and C respectively. $C_{cost}$ represent the minimum cost among the three paths, $C_{cost}^A$, $C_{cost}^B$, $C_{cost}^C$ represent the cost of paths A, B and C, respectively.

Through the matrix operation mentioned above, the transformation from a multi-path problem to a conventional VRPTW problem was realized, which provided a necessary preparation for solving the problem by using CPLEX in Java.

## 2.3. Comprehensive Calculation Model of Fuel Emissions

This paper employed the comprehensive modal emissions model (CMEM) (Demir et al. [5] and Koç et al. [6]) to assess vehicle fuel consumption and emissions levels. In a certain time $t$, the load of the vehicle is f, and the vehicle travels $d$ kilometers at a constant speed v, so the fuel consumption of the vehicle can be calculated as:

$$F(t,v,f) = \lambda k N_e V t + \lambda \gamma \beta v^3 t + \lambda \gamma \alpha d(\mu + f) \tag{5}$$

Therefore,

$$F\left(\frac{dis\tan ce}{speed}, speed, load\right) = \lambda\left(k N_e V \frac{dis\tan ce}{speed} + \gamma\beta dis\tan ce(speed)^2 + \lambda\alpha(\mu + load)dis\tan ce\right) \tag{6}$$

where $\lambda = \varepsilon/(k\psi) = 1/(1000\varepsilon\omega)$, $\alpha = g\sin(\phi) + gC_\gamma\cos(\phi)$, $\beta = 0.5C_d A_f \rho$.

As seen in Table 1, this paper describes a parameter based on a typical light-duty vehicle and its corresponding values. In order to adapt to the actual road conditions in China, some constants and road-related values in the model of Franceschetti et al. were adjusted. The comprehensive emissions

model consists of three parts, namely an engine module, speed module and weight module. The CMEM model clearly indicates that the fuel consumption rate is related to the vehicle's speed and load.

**Table 1.** The comprehensive modal emissions model (CMEM) parameters.

| Category | Parameter | Description | Value |
|---|---|---|---|
| Related parameters of vehicle | $K$ | Engine friction coefficient | 0.2 |
| | $N_e$ | Engine speed | 38.33 |
| | $V$ | Engine displacement | 4.7 |
| | $A_f$ | The area on the front of the vehicle | 5.03 |
| | $\mu$ | Curb weight of the vehicle | 3850 |
| | $\varepsilon$ | Vehicle traction efficiency | 0.4 |
| | $\omega$ | Diesel engine efficiency | 0.9 |
| Related parameters of road | $\phi$ | Road angle | 0 |
| | $C_\gamma$ | Aerodynamic rotational resistance | 0.01 |
| | $\xi$ | Diesel heating value | 1 |
| Others(constant) | $\rho$ | Air density | 1.2041 |
| | $\psi$ | Conversion factor from gram to liter | 737 |
| | $g$ | Gravity constant | 9.81 |
| | $C_d$ | Aerodynamic pull | 0.7 |

## 3. Establishing a Mathematical Model

### 3.1. Symbol Description

In order to describe the model, the following symbols are defined:

$G$: indicates the actual traffic network;

$N$: indicates the collection of customer nodes and distribution centers;

$A$: indicates the collection of road segments in route formulation;

$r_{i,j}$: indicates the number of edges between two nodes;

$p_{ij}$: indicates the set of paths that the arc $(i, j)$ has;

$P_{ij}^P$: indicates that one path is selected from the path set $p_{ij}$;

$K$: indicates the number of vehicles;

$Q$: indicates the capacity of one vehicle;

$f_{ij}$: indicates the load of the vehicle on the arc $(i, j)$;

$f_{ij}^p$: indicates the vehicle load in $P_{ij}^P$;

$q_i$: indicates the demand of the customer node $i$;

$s$: indicates the travel speed of the vehicle;

$C^d$: indicates the depreciation cost per vehicle;

$w$: indicates the driver's salary, which is proportional to the driving time;

$d_{ij}^p$: indicates the length of the road $P_{ij}^P$;

$\lambda_j$: indicates the vehicle technical parameters, $\lambda_j = \xi_j / k_j \psi_j, j \in \{1, 2\}$;

$\mu_j$: indicates the vehicle technical parameters, i.e., the vehicle equipment quality;

$Q_j$: indicates the vehicle technical parameters, where $Q_j = \gamma_j \partial_j, j \in \{1, 2\}$;

$v_j$: indicates the vehicle technical parameters, where $v_j = \gamma_j \beta_j, j \in \{1, 2\}$;

$\omega_j$: indicates vehicle technical parameters, where $\omega_j = k_j N_e^j V_j, j \in \{1, 2\}$;

$e$: indicates the price of fuel per liter;

$T$: indicates the service time of the vehicle at the customer node;

$t_{ij}$: indicates the travel time from $i$ to $j$.

$(E, L)$: indicates the time window of the customer, where $EE_i$ is the earliest service time of the vehicle, $E_i$ is the earliest service time of vehicle standard, $LL_i$ is the latest service time for the vehicle, $L_i$ is the latest for the vehicle standard Service time, $S_i$ is vehicle arrival time; if the vehicle arrives at a node $i$ within a range of $(EE_i, E_i)$ or $(L_i, LL_i)$, it needs to pay the additional cost because of non-compliance

with the service contract. If the vehicle is serviced before $EE_i$ or after $LL_i$, it will be rejected for exceeding the time allowed by the customer.

### 3.2. Establishing a Minimum Cost Mathematical Model

Under a deterministic traffic network, candidate paths $p_{ij}$ belonging to each arc $A^c$ are first processed according to the statistical analysis of the data of the expected traffic network. Next, the planning of the vehicle travel path is carried out in order to minimize the total cost.

Firstly, the decision variables are defined as follows:

$$x_{ij} = \begin{cases} 1 & \textit{if the selected route is the best route in arc}(i,\ j) \\ 0 & \textit{others} \end{cases}$$

$$x_{ij}^p = \begin{cases} 1 & \begin{aligned} &\textit{if the vehicle is driving on arc}(i,\ j), \\ &\quad \textit{selecting a path in the road set} \end{aligned} \\ 0 & \textit{others} \end{cases}$$

$$f_{ij} = \textit{The load of the vehicle via arc}(i, j)$$

$$f_{ij}^p = \textit{The load of the vehicle on the road } \rho$$

Secondly, the penalty function needs to be defined as follows:

$$u_i(S_i) \begin{cases} a_i(E_i - S_i) & \textit{if } EE_i < S_i < E_i \\ 0 & \textit{if } E_i < S_i < L_i \\ b_i(S_i - L_i) & \textit{if } L_i < S_i < LL_i \\ +\infty & \textit{if } S_i < EE_i \textit{ or } S_i > LL_i \end{cases}$$

$E_0$ indicates the departure time from the distribution center. Since the early penalty cost is generally less than the late penalty cost, the time window here is asymmetrical, i.e., $a_i \neq b_i$.

Then, the following mathematical model is formulated:

$$\begin{aligned} MinZ \quad = \quad & \sum_{(i,j)\in A}\sum_{p\in P}\left(\lambda\left(\omega\left(d_{ij}^p/s\right)X_{ij}^p + vd_{ij}^p s^2 X_{ij}^p + Q\left(\mu X_{ij}^p + f_{ij}^p\right)d_{ij}^p\right)\right)e \\ & + \sum_{(i,j)\in A}\sum_{p\in P}\left(d_{ij}^p/s\right)X_{ij}^p w + c^d \sum_{(i,j)\in A}\sum_{p\in P}d_{ij}^p x_{ij}^p + \sum_{(i,j)\in A}\sum_{p\in P}\mu_i(S_i) \end{aligned} \tag{7}$$

s.t.

$$\sum_{j\in N} x_{jo} \leq K \tag{8}$$

$$\sum_{j\in N} x_{oj} \leq K \tag{9}$$

$$x_{ij} \in \{0,1\} \quad \forall(i,j) \in A \tag{10}$$

$$x_{ij}^p \in \{0,1\} \quad \forall(i,j) \in A. \forall p \in P \tag{11}$$

$$f_{ij} > 0 \qquad \forall(i,j) \in A \tag{12}$$

$$f_{ij}^p > 0 \quad \forall(i,j) \in A. \forall p \in P \tag{13}$$

$$\sum_{j\in N} x_{ij} = 1 \quad \forall j \in N\backslash\{0\} \tag{14}$$

$$\sum_{i \in N} x_{ij} = 1 \quad \forall j \in N\backslash\{0\} \tag{15}$$

$$\sum_{i \in N} f_{ij} - \sum_{k \in N} f_{jk} = q_j \quad \forall j \in N\backslash\{0\} \tag{16}$$

$$q_j x_{ij} \leq f_{ij} \leq (Q - q) x_{ij} \quad \forall(i, j) \in A \tag{17}$$

$$\sum_{p \in P} x_{ij}^p = x_{ij} \quad \forall(i, j) \in A \tag{18}$$

$$\sum_{p \in P} f_{ij}^p = f_{ij} \quad \forall(i, j) \in A \tag{19}$$

$$q_j x_{ij}^p \leq f_{ij}^p \leq (Q - q) x_{ij}^p \quad \forall(i, j) \in A. \forall p \in P \tag{20}$$

$$EE_j \leq S_i + T_i + t_{ij} \leq LL_j \quad i, j = 1 \ldots n \tag{21}$$

In the objective function, Equation (7) concerns the minimization of the total fuel cost, driver wage, vehicle depreciation expense related to travel distance and time-window penalty cost, respectively. Constraints (8) and (9) ensure that the number of vehicles used cannot exceed the number of available vehicles *K*, and Constraints (10)–(13) are variable specification constraints. Constraints (14)–(17) are the standard constraints for a double-index single commodity flow model in vehicle routing problems with limited capacity, the first two of which (Constraints (14) and (15)) are conservation constraints of vehicle flow. Constraint (16) is the conservation constraint of vehicle commodity flow, and constraint (17) guarantees that the loaded cargo does not exceed the capacity of the vehicle. Constraint (18) implies that the vehicle can select exactly one of the paths under the customer connection arc $(i, j) \in A$. Constraints (19) and (20) also ensure that the goods are transported along a path, and that the loaded cargo does not exceed the capacity of the vehicle if it travels over an arc $(i, j) \in A$. Moreover, constraint (21) is a time-window constraint.

## 4. Scenario Analysis

In this Section, the distribution of JD logistics in the main urban area of Chongqing was selected as the research object. Firstly, related path data were processed by the matrix operation, and then the problem was solved by using CPLEX functions (version 12.6, IBM, New York, NY, USA, 2015) and MATLAB APIs (version 2016, Mathworks, Natick, MA, USA, 2016) in Java.

### 4.1. Scenario Description

In order to test the applicability of our model and algorithm, we took JD distribution in the main city of Chongqing as an example and analyzed the effect of the algorithm in the actual distribution environment. Here, distribution maps of JD distribution centers and customer locations were obtained through Baidu maps, the map marking tool and other tools, which are shown in Figure 2.

As shown in Figure 2, there were 10 customer nodes and one distribution center in the scenario of JD. In the scenario, the distribution center needed to complete the relevant delivery within the specified time because each customer had its own delivery time window. Meanwhile, the main area of Chongqing has an advanced transportation network, which provided JD with more route choices. Therefore, the distribution center needed to select the optimal travel route to complete the distribution operation in consideration of the delivery time window of each customer node, so as to improve the distribution efficiency of JD logistics.

### 4.2. Study Solution and Result Analysis

This paper mainly solved the problem using CPLEX, mathematical optimization software from IBM, which provides a flexible and high-performance optimization program with the advantages of fast speed and exact problem solving capabilities. However, as the integrated development environment

(IDE) of CPLEX is not friendly enough, the ability to solve complex vehicle routing problems was insufficient, and the problem description was not detailed and accurate enough. Therefore, in this article, we employed Java to invoke CPLEX to avoid the disadvantages of CPLEX IDE and better solve the multi-path vehicle routing problem. In summary, since using Java to invoke CPLEX was very suitable for solving this problem, we used Java to invoke CPLEX v12.6.1 to solve the VRPTW problem with 11 nodes.

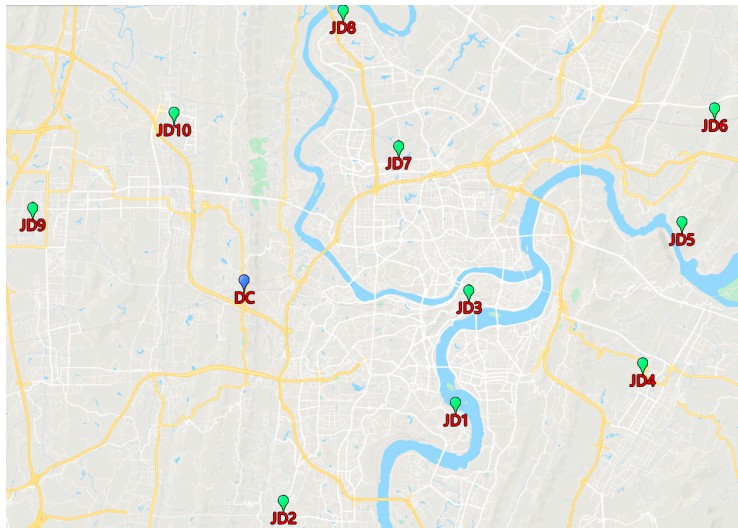

**Figure 2.** The distribution map of Jingdong (JD) distribution center and customer nodes.

By contrast with the traditional VRPTW problem, a hard time window has strict vehicle departure time and waiting time requirements in the customer node.

In this paper, the vehicle routing problem based on multi-paths was studied. Moreover, the addition of a hard time window undoubtedly greatly reduced the candidate set of paths, which eliminated some high-quality road sections from the datasets and significantly changed the vehicle path formulation. In this case, 10 nodes were distributed by multiple vehicles, each carrying a load of 5 metric tons. Vehicles were required to service customers within a specified time window, otherwise, a penalty cost was incurred due to early or late arrival. However, since the problem involved in this paper was a multi-path vehicle routing problem, the addition of a hard time windows undoubtedly greatly compressed the path selectable range, which enabled JD logistics' operating nodes to be distributed in different locations according to the demand of 10 respective branches (1.9, 1.7, 1.8, 2.1, 1.6, 2.4, 2.2, 1.8, 2.4, 1.9).

### 4.2.1. Solution Analysis

In this paper, an Intel i5 1.9 GHz central processing unit (CPU) computer with 4 GB RAM was used to solve mathematical programming (7)–(21) through CPLEX v12.6.1, and 3156 consecutive Numbers, 1987 binary variables and 4156 constraints were obtained. In an urban distribution, different requirements are put forward for logistics companies due to different properties of goods and different needs of customers. For example, more attention is paid to minimize costs in large-scale distribution, while more attention is paid to timeliness in drug distribution. Therefore, in order to make this study more suitable for the actual needs, four different objective functions were designed, including the shortest distance objective function, the shortest time objective function, the minimum fuel consumption objective function and the minimum cost objective function, as shown in Formulas (22) and (23).

In order to obtain the travel solution with the shortest distance, the following function was designed in this paper:

$$Minmize \sum_{(i,j)\in A} \sum_{p\in P} d_{ij}^p X_{ij}^p \tag{22}$$

In order to obtain the travel solution with the shortest time, the following function was designed in this paper:

$$Minmize \sum_{(i,j)\in A} \sum_{p\in P} \left(d_{ij}^p / s_{ij}^p\right) X_{ij}^p \tag{23}$$

In order to obtain the travel solution with the least fuel consumption and the minimum cost, we employed Equations (6) and (7) as the objective functions. Different models use the same constraint conditions, i.e., Constraints (18)–(21). Table 2 clearly illustrates the differences among the various models.

**Table 2.** Differences among various models.

| Model Variable | Travel Distance | Vehicle Speed or Road Congestion | Vehicle Load | Fuel Price or Driver's Salary |
|---|:---:|:---:|:---:|:---:|
| Minimum distance | √ | | | |
| Shortest time | √ | √ | | |
| Minimal fuel consumption | √ | √ | √ | |
| Lowest cost | √ | √ | √ | √ |

Through the analysis above, it can be seen that the variables covered by different models were significantly different. Therefore, Constraints (8)–(21) were attached to the four different objective functions, and then CPLEX and MATLAB were invoked by Java to solve the problem. The following results were obtained, as shown in Table 3.

**Table 3.** Comparisons of scenarios of 11 nodes with different objective functions.

| Path and Key Index | Minimum Distance | Minimum Time | Minimum Fuel | Minimum Cost |
|---|:---:|:---:|:---:|:---:|
| | 0-9-10-0 | 0-3-1-0 | 0-2-10-0 | 0-8-7-0 |
| | 0-8-7-0 | 0-4-9-0 | 0-4-9-0 | 0-6-2-0 |
| Tour plan | 0-4-3-0 | 0-6-7-0 | 0-6-5-0 | 0-9-5-0 |
| | 0-1-2-0 | 0-8-10-0 | 0-8-7-0 | 0-4-10-0 |
| | 0-6-5-0 | 0-2-5-0 | 0-3-1-0 | 0-3-1-0 |
| Total traveled distance (km) | 319.5 | 325.55 | 323.65 | 323.45 |
| Total traveled time (minute) | 424 | 420 | 427 | 429 |
| Total fuel (liter) | 42.63 | 44.95 | 42.61 | 43.84 |
| Total cost (€) | 552.18 | 559.33 | 555.64 | 544.64 |

From the analysis in Table 3, it can be seen that vehicles had different path choices in different objective functions. By assigning the same Constraints (11)–(24) to the objective functions (25), (26), (6), and (7)–(10), we obtained the routing formulation of the optimal time, distance, fuel consumption, and cost, respectively. At the same time, four main performance indicators under a different routing formulation were obtained by MATLAB. It was found that there were significant differences among the four indicators under a different route formulation, and the indicators that were consistent with the objective function gave the best results.

### 4.2.2. Sensitivity Analysis

Path Selection Sensitivity Analysis

Based on the above analysis, it was determined that the path selection between customer nodes in the study of this problem had a significant impact on the formulation of vehicle paths and the

four key performance indicators. Therefore, sensitivity analysis was performed on the entire model in the next step when the customer's path selection changed. We reduced the number of paths between nodes to one when the paths were adjusted. In this paper, the path information in the case of label K = A was directly selected, and the calculated result was compared with the optimal path resulting from the processing of the data. Finally, the path change was found, which impacted the four key performance indicators. In the sensitivity analysis of this paper, Java-invoked CPLEX was also employed. The specific results are shown in Table 4.

**Table 4.** Scenarios of 11 nodes with different objective functions when K = A.

| Path and Key Indicators | Minimum Distance | Shortest Time | Minimal Fuel Consumption | Lowest Cost |
|---|---|---|---|---|
| | 0-2-10-0 | 0-9-5-0 | 0-3-1-0 | 0-9-10-0 |
| | 0-4-9-0 | 0-4-3-0 | 0-4-9-0 | 0-4-3-0 |
| Tour plan | 0-6-5-0 | 0-6-10-0 | 0-2-5-0 | 0-6-5-0 |
| | 0-8-7-0 | 0-8-7-0 | 0-6-10-0 | 0-8-7-0 |
| | 0-1-3-0 | 0-1-2-0 | 0-8-7-0 | 0-2-1-0 |
| Total traveled distance (km) | 323.65 | 326.5 | 354 | 335 |
| Total traveled time (minute) | 451 | 443 | 454 | 450 |
| Total fuel (liter) | 52.28 | 55.17 | 51.62 | 52.15 |
| Total cost (€) | 598.16 | 611.16 | 603.13 | 595.64 |

Table 4 shows the four key performance indicators under different objective functions when K = A. By comparing this data with the data in Table 3, Figure 3 was obtained.

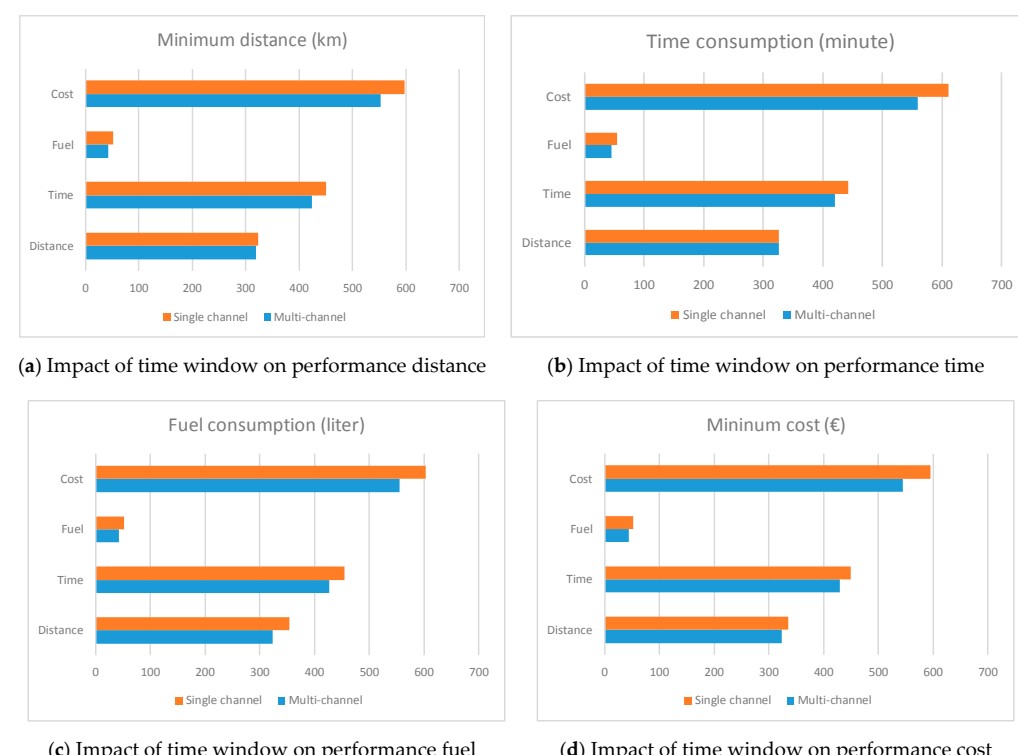

(**a**) Impact of time window on performance distance

(**b**) Impact of time window on performance time

(**c**) Impact of time window on performance fuel

(**d**) Impact of time window on performance cost

**Figure 3.** Channel selection impact on performance indicators.

Through analysis of the results shown in Figure 3, it was seen that without considering multi-channels, the results under different objective functions were not perfect. Except for the small difference in performance between the benchmark instance and K = A instance under the minimum fuel consumption indicator, the performance of the main indicators of K = A under the other three performance indicators was not satisfactory. In the two key indicators of time and cost, in particular, the routing formulation without considering multi-paths consumed more time and cost

than the benchmark instance. Such results effectively indicate that considering multi-channels has a significant impact on vehicle routing formulation.

Time-Window Sensitivity Analysis

In order to further understand the impact of time windows on multi-path vehicle routing problems, the changes of four key indicators without time-window constraints were analyzed. Multi-path traffic conditions were also considered, but the time-window constraint from Section 4.2.1 was removed; the results are shown in Table 5.

**Table 5.** Examples of different objective functions with no time-window constraints at 11 nodes.

| Path and Key Indicators | Minimum Distance | Shortest Time | Minimal Fuel Consumption | Lowest Cost |
| --- | --- | --- | --- | --- |
| | 0-9-10-0 | 0-3-4-0 | 0-2-10-0 | 0-8-2-0 |
| | 0-8-7-0 | 0-5-9-0 | 0-4-9-0 | 0-7-6-0 |
| Tour plan | 0-4-3-0 | 0-6-7-0 | 0-6-8-0 | 0-9-5-0 |
| | 0-1-2-0 | 0-8-10-0 | 0-5-7-0 | 0-4-10-0 |
| | 0-6-5-0 | 0-2-1-0 | 0-3-1-0 | 0-3-1-0 |
| Total traveled distance (km) | 314.6 | 318.89 | 320.25 | 321.35 |
| Total traveled time (minute) | 406 | 411 | 401 | 405 |
| Total fuel (liter) | 41.52 | 42.87 | 41.23 | 40.96 |
| Total cost (€) | 501.98 | 508.56 | 512.31 | 507.49 |

Table 5 shows four key performance indicators under different objective functions without time-window constraints. By comparing the data in Table 5 with the data in Table 3, we obtained the results shown in Figure 4.

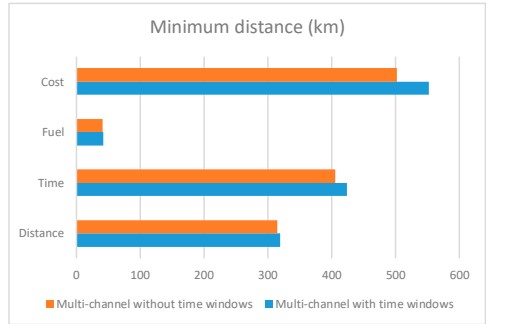

**(a)** Impact of time window on performance distance

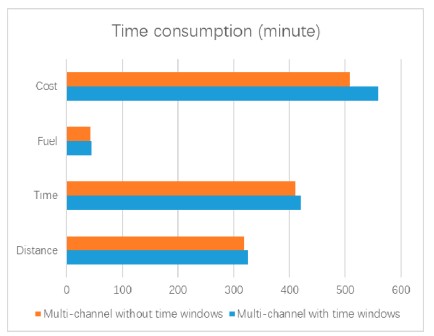
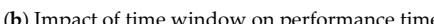

**(b)** Impact of time window on performance time

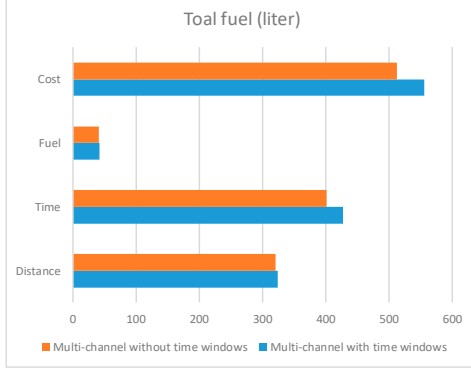

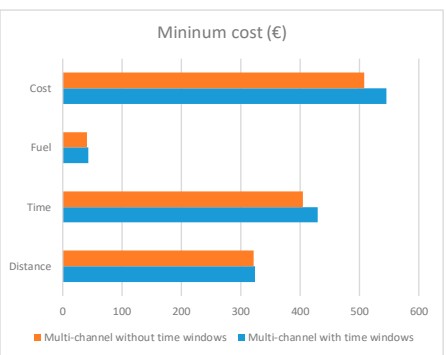

**Figure 4.** Impact of time window on performance indicators.

Through the analysis of Figure 4, it can be determined that the influence of the time window on the time index was the largest. In the absence of time-window constraints, time indicators showed better results, and other indicators were also optimized to varying degrees. This was mainly due to the further increase in the selectivity of the route after eliminating the time-window constraint, which also increased the possibility of choosing a better route during driving.

## 5. Conclusions

In this paper, we analyzed the multi-path vehicle routing problem with time windows and established a mathematical model to solve the problem with CPLEX and other tools. In this process, we established four different objective functions and then formulated four different vehicle paths to meet the distribution needs of different logistics companies. Meanwhile, we also calculated four key performance indicators under different vehicle paths and then evaluated the pros and cons of each. Therefore, this study has important practical significance.

(1) In order to evaluate the advantages of considering multi-paths, sensitivity analysis of paths was also carried out. The analysis shows that the three performance indicators of vehicle path, time, cost and distance all more or less decreased when considering multiple channels, and at the same time, it also shows obvious advantages in fuel consumption.

(2) Numerical results of the analyses show that the traditional minimum objectives in distance and time cannot guarantee the minimum fuel consumption and cost.

(3) The time-window constraint has a significant impact on the results of multi-path vehicle routing problems; the time window in particular greatly affects the vehicle path selection space. This makes the time window-less constraint increase the probability of vehicles choosing a better path, and thus has the opportunity for better search results.

This paper is only an exploratory study of a multi-channel vehicle routing problems, so the following cases may be further studied.

(1) The research in this paper is based on a static environment. However, a dynamic multi-path vehicle routing problem has not been carried out, which is of more practical significance.

(2) In this paper we only studied the path formulation under different objective functions, and further research on multi-objective multi-path vehicle routing problems is needed.

(3) The research in this paper is mainly based on CPLEX, which means the solution scale is limited. Therefore, future research directions should include new heuristic algorithms to solve large-scale multi-path vehicle routing problems.

**Author Contributions:** Conceptualization, W.W.; methodology, X.G. (Xianlong Ge); software, X.G. (Xianlong Ge); formal analysis, W.W.; resources, W.W.; writing—original draft preparation, X.G. (Xiaobo Ge); writing—review and editing, W.W. All authors have read and agreed to the published version of the manuscript.

**Funding:** This research was funded by National Social Science Foundation of China, grant number 19CGL041.

**Conflicts of Interest:** The authors declare no conflict of interest.

## Appendix A

**Table A1.** Abbreviations.

| No. | Abbr. | Description |
|-----|-------|-------------|
| 1 | PRP | pollution emissions problem |
| 2 | FIFO | first-in and first-out |
| 3 | TS | tabu search |
| 4 | CMEM | integrated modal emissions model |
| 5 | TDVRP | time-dependent vehicle routing problems |
| 6 | VRPTW | vehicle routing problems with time windows |
| 7 | OP | optional path |

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
