# Peer review of "A Path-Based Selection Solution Approach for the Low Carbon Vehicle Routing Problem with a Time-Window Constraint"

_applsci, doi:10.3390/app10041489_

Round 1

Reviewer 1 Report

The paper describes the mathematical formulation of an approach for the solution of the classical vehicle routing problem where multiple paths and multiple objective functions are considered. Such topic is faced through mixed integer linear programming and some preliminary simulation results are presented and discussed.

The topic of the paper is interesting and is in line with the Journal aims and scope.

The paper is interesting and might have a practical relevance to many readers, as vehicle routing is a general issue, which actually does not refer only to logistic and traffic management. It can also have many industrial applications.

The introductory section contains a very accurate and comprehensive analysis of the state of the art, which duly highlights the elements of novelty and strength of the proposed approach.

The proposed approach is interesting and technically sound, although it is only a preliminary study. The authors correctly highlight in the conclusions that further investigations and developments are needed before it can be practically applied, being basically the major limitations represented by the consideration of a static environment and the use of CPLEX. Nonetheless the paper overall holds some merit and some aspects which would deserve publication. However, the paper contains a number of major flaws, which prevent its publication in its present stage.

The use of the symbols is often not correct: as a general rule, in order to make the paper readable, especially with complex mathematical formulations, the correspondence between symbols and variables should be bi-univocal. This means that each variable must be indicated with one symbol only (see here, for instance, the use of F and Fuel in Subsection 2.3) and the same symbol cannot be used to indicate two different variables (see here, for instance, the different meanings of the variable m in Subsections 2.3 and 3.1).

Subsection2.2 is unclear and needs to be revised: Formulas 1-4 contain symbols, which are not defined beforehand. Moreover, such formulas are not enough justified in the text.

In Subsection 2.3 Formulas 5 and 6 contains symbols, which are not defined, such as F, Fuel, a and Af. Moreover all the formulas need to be numbered.

The sensitivity analysis described in Section 4.2.2 is a strong point of the described research work, but the explanations provided in the first paragraph of this Subsection (i.e. the paragraph placed before Table 4) is unclear and insufficient in order to described the developed analysis. It needs to be clarified and extended.

The English language absolutely needs improvement, as there are many incorrect and unclear sentences, which sometimes even hamper the full understanding of the paragraphs and, in general, make the paper hard to read. A few examples:

Page 1 rows 36-37: “this makes the vehicle always choosing…”; Page 3 rows 104-105: “w3,5=2 indicates that there are two edges between nodes (2,3), in other words existing two roads connection between them”. (and the numbers are also wrong here, as the considered nodes should be 3 and 5; Page 3 row 114: “…the number of paths between any two nodes may be many choices…”; Page 4 row 134: “… which provides a necessary prepare for solving…”; Page 10, row 341-342:”…it is necessary for further research”: I think that the meaning actually is “further research is needed”

There are many other examples throughout the paper of English language mistakes, therefore I suggest a proofreading done by a native English speaker.

As a minor formal remark, some of the formulas do not comply with the format provided by the journal.

Finally, I suggest to add a list of acronyms at the end of the paper.

Author Response

Point 1: The use of the symbols is often not correct: as a general rule, in order to make the paper readable, especially with complex mathematical formulations, the correspondence between symbols and variables should be bi-univocal. This means that each variable must be indicated with one symbol only (see here, for instance, the use of F and Fuel in Subsection 2.3) and the same symbol cannot be used to indicate two different variables (see here, for instance, the different meanings of the variable m in Subsections 2.3 and 3.1).

Response 1: Thanks for the valuable comment and suggestion. According to your valuable suggestions, we have carefully checked all symbols and revised the mistakes.

Point 2: Subsection2.2 is unclear and needs to be revised: Formulas 1-4 contain symbols, which are not defined beforehand. Moreover, such formulas are not enough justified in the text.

Response 2: Thanks for the valuable comment and suggestion. We updated the manuscript by modifying Formulas 1-4.

Point 3: In Subsection 2.3 Formulas 5 and 6 contains symbols, which are not defined, such as F, Fuel, and Af. Moreover all the formulas need to be numbered.

Response 3: Thanks for the valuable comment. We have carefully checked Formulas 5 and 6 and its description, then three problems were addressed.

Point 4: The sensitivity analysis described in Section 4.2.2 is a strong point of the described research work, but the explanations provided in the first paragraph of this Subsection (i.e. the paragraph placed before Table 4) is unclear and insufficient in order to described the developed analysis. It needs to be clarified and extended.

Response 4: We updated the manuscript by modifying section 4.2.2. The time window sensitivity analysis is added in Section 4.2.2. The detailed revision is shown in the new section of “Time window sensitivity analysis”, and marked by blue color.

Point 5: The English language absolutely needs improvement, as there are many incorrect and unclear sentences, which sometimes even hamper the full understanding of the paragraphs and, in general, make the paper hard to read.

Response 5: We are grateful to your suggestion that improved our manuscript greatly. To improve the quality of communication, we asked for the professional language review of some native English speakers and MDPI English editing, and carefully checked and modified the typos, grammatical and other mistakes of this paper.

Point 6: Page 1 rows 36-37: “this makes the vehicle always choosing…”; Page 3 rows 104-105: “w3,5=2 indicates that there are two edges between nodes (2,3), in other words existing two roads connection between them”. (and the numbers are also wrong here, as the considered nodes should be 3 and 5; Page 3 row 114: “…the number of paths between any two nodes may be many choices…”; Page 4 row 134: “… which provides a necessary prepare for solving…”; Page 10, row 341-342:”it is necessary for further research”: I think that the meaning actually is “further research is needed”

Response 6: Your suggestions are so kind to point out the mistakes of this work, We are really sorry for the careless mistakes. According to your valuable suggestions, we have carefully revised your mentioned mistakes, and we have also carefully recheck the whole paper to revise other mistakes.

Point 7: There are many other examples throughout the paper of English language mistakes, therefore I suggest a proofreading done by a native English speaker.

Response 7: We are grateful to your suggestion that improved our manuscript greatly. To improve the quality of communication, we asked for the professional language review of some native English speakers and MDPI English editing, and carefully checked and modified the typos, grammatical and other mistakes of this paper.

Point 8: As a minor formal remark, some of the formulas do not comply with the format provided by the journal.

Response 8: We are grateful to your suggestions that improved our manuscript greatly. We checked and modified the format of the formulas.

Point 9: Finally, I suggest to add a list of acronyms at the end of the paper.

Response 9: Thanks for the valuable comment and suggestion. An appendix of all abbreviations mentioned in this paper was added before the references.

Reviewer 2 Report

The paper is well written and easy to follow. The Introduction is comprehensive and it also includes some clues related to the research literature. Section 2, the Problem Description, presents the needed information in order to understand the proposed approach. The analyzed problem looks complex and seems to be solved on a particular case: JD Logistics. I think that starting from this case, the model should prove its applicability in other situations. Please try to generalize it more. Also, please increase the quality of Figure 3.

Minor corrections:

please check the information in rows 104-105: wouldn't it be between nodes 3 and 5 instead of 2 and 3? also, in row 105 and 109, please replace fig.1 with Figure 1.

Author Response

Point 1: The paper is well written and easy to follow. The Introduction is comprehensive and it also includes some clues related to the research literature. Section 2, the Problem Description, presents the needed information in order to understand the proposed approach. The analyzed problem looks complex and seems to be solved on a particular case: JD Logistics. I think that starting from this case, the model should prove its applicability in other situations. Please try to generalize it more. Also, please increase the quality of Figure 3.

Response 1: Thanks for the valuable comment and suggestion. According to your valuable suggestions, we have improved my model to some extend and provided a high-resolution image for Figure 3.

Point 2: please check the information in rows 104-105: wouldn't it be between nodes 3 and 5 instead of 2 and 3? also, in row 105 and 109, please replace fig.1 with Figure 1.

Response 2: Thanks for the valuable comment and suggestion. We have carefully corrected the errors in rows 104,105 and 109. Meanwhile, all abbreviations “fig.” have been replaced by “Figure”.

Reviewer 3 Report

This paper treats the optimal time dependant path problem for the traffic/logistics problem related to fuel consumption. Authors use CMEM comprehensive emission model for the fuel cost function where three main variables are considered namely, load, distance and average speed of the vehicle on the road. Path/road parameters are modelled via some statistical data analysis. Different linear cost functions with linear constraints are solved with IBM-CPLEX solver and compared with Matlab solver.

The topic is interesting for a publication and method looks sound. The optimality condition is investigated and compared with experimental data. Experimental evidence confirm the optimal decision obtained with proposed mathematical models. In the end, the sensitivity analysis is performed by the CPLEX tool revealing new insight into the vehicle/path parameters nature.

Results are based on statistical analysis and only averages are taken into account. In the sensitivity analysis, the most tricky parameters are identified. Probably, it is a good idea to continue with the investigation on the impact of the uncertainty of the parameters on the optimality of the solution.

Author Response

Point 1: In the sensitivity analysis, the most tricky parameters are identified. Probably, it is a good idea to continue with the investigation on the impact of the uncertainty of the parameters on the optimality of the solution.

Response 1: Thanks for the valuable comment and suggestion. We added time window sensitivity analysis in Section 4.2.2. The detailed revision is shown in the new section of “Time window sensitivity analysis”, and marked by blue color.

Round 2

Reviewer 1 Report

The paper describes the mathematical formulation of an approach for the solution of the classical vehicle routing problem where multiple paths and multiple objective functions are considered. Such topic is faced through mixed integer linear programming and some preliminary simulation results are presented and discussed.

The topic of the paper is interesting and is in line with the Journal aims and scope.

The authors carefully amended the paper according to the suggestions they received in the revew report. The paper has been drastically improved and now is suitable to publication.